# Treatment of Vitamin D Deficiency with Calcifediol: Efficacy and Safety Profile and Predictability of Efficacy

**DOI:** 10.3390/nu14091943

**Published:** 2022-05-05

**Authors:** Jose-Luis Pérez-Castrillon, Ricardo Usategui-Martín, Pawel Pludowski

**Affiliations:** 1Servicio de Medicina Interna, Hospital Universitario Río Hortega, University of Valladolid, 47012 Valladolid, Spain; 2Department of Cell Biology, Histology and Pharmacology, Faculty of Medicine, University of Valladolid, 47003 Valladolid, Spain; ricardo.usategui@uva.es; 3IOBA, University of Valladolid, 47011 Valladolid, Spain; 4Department of Biochemistry, Radioimmunology and Experimental Medicine, The Children’s Memorial Health Institute, 04730 Warsaw, Poland; p.pludowski@ipczd.pl

**Keywords:** calcifediol, cholecalciferol, vitamin D deficiency, efficacy, toxicity, predictability

## Abstract

Calcifediol (25-OH-vitamin D3) is the prohormone of the vitamin D endocrine system. It is used to prevent and treat vitamin D deficiency. Calcifediol, as well as cholecalciferol (vitamin D3), is efficient and safe in the general population, although calcifediol has certain advantages over cholecalciferol, such as its rapid onset of action and greater potency. This review analyzed studies comparing the efficacy and safety of both calcifediol and cholecalciferol drugs in the short and long term (>6 months). Calcifediol was found to be more efficacious, with no increase in toxicity. We also assessed the predictability of both molecules. A 25OHD increase depends on the dose and frequency of calcifediol administration. In contrast, after cholecalciferol administration, 25OHD increase depends on more factors than dose and frequency of administration, also phenotypic aspects (such as obesity and malabsorption), and genotypic factors impacts in this increase.

## 1. Introduction

Vitamin D3 is the nutrient of the vitamin D endocrine system (VDES). It is derived from the steroid group that is synthesized endogenously from 7-dehydrocholesterol (provitamin D3), which is converted into 7-dehydrocholecalciferol in the upper layers of the skin by the action of ultraviolet radiation [1]. Vitamin D is also derived from diet and may be of animal origin (D3; cholecalciferol) or vegetable origin (D2; ergocalciferol). Cholecalciferol is bound to a transport protein (vitamin D binding protein [DBP]) and reaches the liver, where it is metabolized by 25-hydroxylases (mainly microsomal CYP2R1, mitochondrial CYP27R1) to 25-hydroxycholecalciferol (calcifediol; calcidiol; 25OHD3), the prohormone and cornerstone of the VDES [2]. Calcifediol binds to DBP and is transported mainly to the kidneys, where it is metabolized by the effect of 1α-hydroxylase (CYP27B1) or 24-hydroxylase (CYP24A1) [3]. The active metabolite of vitamin D3 is 1,25 dihydroxycholecalciferol, or calcitriol, whose synthesis is endocrinologically controlled, stimulated by parathyroid hormone (PTH), and inhibited by fibroblast growth factor 23 (FGF23) [4]. 24,25-dihydroxycholecalciferol production is inhibited by PTH and increased by FGF23 [1,4].

Vitamin D deficiency—measured as levels of 25OHD—is common worldwide. The potential adverse effects of poor vitamin D status are of concern for public health [5]. Dietary and pharmacological supplementation probably have no additional effects when ultraviolet radiation maintains vitamin D status within an adequate range through its endogenous component [6]. The administration of fortified foods in persons with low 25OHD levels is an alternative for achieving recommended and/or desirable levels [7]. However, the most commonly used options involve supplementation with vitamin D2 (ergocalciferol), vitamin D3 (cholecalciferol), and treatment with calcifediol, while calcitriol is indicated in chronic kidney failure and hypo- and pseudohypoparathyroidism [8]. While both cholecalciferol and calcifediol increase 25OHD levels, calcifediol is more potent, rapidly reaching desirable values [9]. In the long term, both drugs behave similarly, although calcifediol, given its pharmacokinetic properties, is preferred in situations such as liver failure, chronic kidney failure, malabsorption, and obesity. This is due to its greater solubility, decreased entrapment in adipose tissue, smaller volume of distribution, avoidance of hepatic metabolism, different mechanism of absorption. The mean half-life is different (Calcidediol, 10–22 days, cholecalciferol 14 days, calcitriol 9–10 h). In addition, it has a greater affinity for the transport protein, which enables more efficient internalization within the megalin-cubilin system [10].

The aim of this review was to compare the short- and long-term efficacy and safety of cholecalciferol and calcifediol. We also assessed the predictability of their action.

### 1.1. Methods

We performed a comprehensive review of the literature through the MEDLINE, PubMed, Web of Science, Scopus, and Embase electronic databases. Potentially relevant articles were sought by using the search terms in combination as Medical Subject Headings (MeSH) terms and text words: “calcidiol trial”, “cholecalciferol trial vs calcidiol”, “calcidiol safety“. In addition, we scanned the reference lists of the retrieved publications to identify additional relevant articles. The search was supplemented using the MedLine option “Related Articles”. No language restrictions were applied. The abstracts for each article were studied to ensure relevance and significance to the review. The safety was evaluated to by the presence of hypercalcaemia, hypercalciuria and nephrolithiasis or by clinical manifestations derived from the presence of hypercalcaemia and vitamin D levels.

### 1.2. Short- and Long-Term Efficacy and Safety of Calcifediol

Studies that analyzed the effect of calcifediol on plasma 25OHD levels were evaluated. Most had a comparator arm with cholecalciferol. Short-term studies compared different doses of calcifediol and showed notable improvements in 25OHD levels. Long-term studies (>6 months) tended to be more heterogeneous in terms of dose and duration of follow-up [11,12].

## 2. Short-Term Efficacy

Two studies on calcifediol lack a cholecalciferol control arm. In their prospective study, Russo et al. [12] administered a dose of 500 µg per month in 18 healthy females aged 24–72 years. The follow-up period was 120 days, and the results showed an increase in 25OHD levels at 7 days, with subsequent stabilization always remaining above baseline values (18.1 ± 12.5 ng/mL, 45.1 ± 31.1 nmol/L) and, in most cases, above 30 ng/mL (74.8 nmol/L). This was associated with a decrease in PTH and bone alkaline phosphatase and unchanged type 1 collagen. In a multicenter, randomized, open-label, 3-arm, parallel-group, comparative phase III study, Minisola et al. [8] assessed the effect of 3 different doses (20 µg/day, 40 µg/day, 125 µg/week) in 87 postmenopausal females with vitamin D deficiency (baseline 25-vitamin D 16.5 ± 7.5 ng/mL, 41.1 ± 18.7 nmol/L). A linear increase in 25OHD concentrations was observed at 14 days and was maintained at 90 days, with a dose-dependent effect. Patients maintained sufficiency (30 ng/mL, 74.8 nmol/L)) with all doses used, which, in no case, exceeded 90 ng/mL (224.6 nmol/L) and were accompanied by reductions in PTH and FGF-23 at 90 days.

Other studies included a control group with cholecalciferol at variable doses. Cashman et al. [13] assessed the effect of 20 µg/day of cholecalciferol versus 7 µg/day or 20 µg/day of calcifediol in 56 patients (31 female and 25 male) aged ≥50 years for 10 weeks in winter. The increase observed was greater for calcifediol than for cholecalciferol, and calcifediol appeared to be 5 times more potent than cholecalciferol. PTH levels decreased in all cases. Bischoff-Ferrari et al. [14] analyzed 20 healthy postmenopausal females in whom 20 µg/day or 140 µg/week of calcifediol was compared with 20 µg/day or 140 µg/week of cholecalciferol. The primary objective was to assess the lower extremity response, blood pressure, and markers of innate immunity. Vitamin D status increased more rapidly and with higher values in patients receiving calcifediol than in those receiving cholecalciferol (69.5 ng/mL (173.4 nmol/L) vs 31 ng/mL (77.3 nmol/L), *p* < 0.0001). PTH values did not differ significantly, probably due to the small sample size. Jetter et al. [11] analyzed the pharmacokinetics of cholecalciferol and calcifediol in 35 healthy women aged 50–70 years divided into 4 arms with different doses (20 µg/day of calcifediol and 20 µg/day of cholecalciferol, 140 µg/week of calcifediol, and 140 µg/week cholecalciferol). Three other arms consisted of a single 140-µg dose of calcifediol and cholecalciferol or a combination of both. All women treated with calcifediol achieved values above 30 ng/mL (74.8 nmol/L), compared with 70% of those treated with cholecalciferol, who also took longer to achieve these values (64.8 days vs 16.8 days). The difference in potency between the 2 treatments was 2- to 3-fold.

Shieh et al. [15] compared the effect of cholecalciferol 60 µg/day versus calcifediol 20 µg/day in 35 patients with vitamin D deficiency. The study lasted 16 weeks. The results were similar to those of previous studies, with a higher increase in 25OHD serum levels (25.5 (63.6 nmol/L) vs 13.8 ng/mL (34.4 nmol/L), *p* < 0.001) and free 25OHD levels (6.6 vs 3.5 pg/mL, *p* < 0.03) in the calcifediol group than in the cholecalciferol group. Vitamin D levels normalized (≥30 ng/mL, 74.8 nmol/L) at 1 month in 87.5% of patients receiving calcifediol compared with 23.1% of those receiving cholecalciferol.

Perez-Castrillón et al. [16] conducted a study whose sample size provided sufficient statistical power to observe differences. The study population comprised 298 postmenopausal women with vitamin D deficiency who received 266 µg /month of calcifediol versus 625 µg /month of cholecalciferol, that is, a 2.5-fold greater dose. The authors reported the results at 1 and 4 months. At 1 month, 13.5% of those who received calcifediol achieved vitamin D sufficiency (30 ng/mL, 74.8 nmol/L) compared with 0% in the cholecalciferol group. At 4 months, the differences were maintained (35% vs 8.2%, *p* < 0.0001). At 16 weeks, the increase in 25OHD was 14.9 ± 8.1 ng/mL (37.1 ± 20.2 nmol/L) vs 9.9 ± 5.7 ng/mL (24.7 ± 14.2 nmol/L) (*p* < 0.0001) for the calcifediol and cholecalciferol groups, respectively. It could be concluded that the difference in potency observed in this study was 3.8 times greater in favor of calcifediol vs cholecalciferol. There were no changes in PTH or markers of bone remodeling.

All short-term studies that compared different doses of calcifediol and cholecalciferol showed that the former increased 25OHD levels faster and by greater amounts [17]. The responses of PTH and markers of bone remodeling were variable, probably because of differences in sample size and variations in the percentages of patients with values >30 ng/mL (74.8 nmol/L). In the only study with sufficient statistical power to observe differences, doses of calcifediol and cholecalciferol were small and may not have been sufficient to inhibit PTH and bone remodeling [16]. Table 1 describes the main studies.

## 3. Long-Term Efficacy

Various studies have analyzed treatment with calcifediol for periods of >6 months, although these are less uniform, and follow-up times and doses are heterogeneous.

The oldest study, which did not include a control group, was a randomized study with the systematic inclusion of all patients treated for hypovitaminosis D in a rheumatology unit [18]. Seventy patients were included (11 male and 59 female, age 70 ± 11 years). All patients received a loading dose of 266 µg of calcifediol weekly for 4 weeks and were subsequently randomized to 3 arms: 266 µg monthly, 266 µg every 3 weeks, and 266 µg every 2 weeks. All patients achieved high 25OHD concentrations, ranging from 59.5 (145.5 nmol/L) to 69.8 ng/mL (174.2 nmol/L) at the third follow-up visit (second and third follow-up visits every 8 ± 4 months), with no associated adverse effects. Follow-up lasted 28 ± 14 months, the longest of all the studies evaluated. Following the same treatment regimen, the authors subsequently conducted a study that included an arm with cholecalciferol [19] in 129 patients (20 male and 109 female) who underwent loading treatment with calcifediol for 4 weeks (266 µg week). The mean 25OHD level achieved was 86 ng/mL (214.6 nmol/L). Patients were then randomized to 2 arms: cholecalciferol 20 μg /day versus calcifediol 266 µg /every 3 weeks. Higher vitamin D values were achieved at 12 months with calcifediol (70 ± 24 ng/mL (174.7 ± 59.9 nmol/L) vs 48 ± 23 ng/mL (119.8 ± 57.4 nmol/L), *p* = 0.001).

Rossini et al. [20] conducted a randomized trial that included 271 women with osteopenia or osteoporosis with associated hypovitaminosis D. Two groups were established: one received 66.5 μg of calcifediol weekly, and the other 20 µg/day of cholecalciferol, with 1 g of calcium administered in both groups. Follow-up was irregular, with a high degree of dropout in patients who received the daily supplements, probably in relation to the added calcium. The 25OHD concentration values at the end of the study were similar in both groups.

Navarro Valverde et al. [21] included 40 osteopenic postmenopausal women with a mean age of 67 years and hypovitaminosis D. Patients were randomized to 4 groups, with similar clinical and demographic characteristics, and received cholecalciferol at 20 µg/day, calcifediol at 20 µg/day, calcifediol at 266 µg/week or calcifediol at 266 µg/2 weeks. The follow-up period was 12 months, and PTH and markers of bone remodeling were determined. Increases in 25OHD concentrations were higher in the group receiving calcifediol, although all groups achieved sufficiency at 6 months. However, the study did not include previous measures with which to draw comparisons. The decrease in PTH was higher in patients who received calcifediol, while the behavior of remodeling markers was irregular. The difference in potency between the 2 supplements was 3- to 5-fold higher in the calcifediol group and more pronounced at higher doses.

An Italian study analyzed patients with hypovitaminosis D (60%) admitted to an acute geriatric unit (77 patients (46 females and 31 males), age ≥75 years) [22]. The patients were randomized to calcifediol 150 µg/week or cholecalciferol 150 µg/week. From 10 days onward, and differences in 25OHD concentrations were observed between the 2 therapies, with calcifediol proving superior, an effect that was maintained at 60 days. At the end of the study, there were no differences between the 2 drugs, although the increase in concentration was higher with calcifediol (19 ng/mL (47.4 nmol/L) vs 16 ng/mL (39.9 nmol/L), *p* = 0.5). PTH decreased similarly in both groups.

Vaes et al. [23] carried out a small, randomized, controlled, double-blind study in patients aged ≥65 years. The control group received 20 µg/day of cholecalciferol, and a further 3 arms were established (5, 10, and 15 µg/day of calcifediol). The study lasted 24 weeks. Stability was reached at 2 weeks, and only the doses of calcifediol 10 and 15 µg/day reached levels of >30 ng/mL (74.8 nmol/L), with all mean values achieved at higher calcifediol levels than in the control group.

A real-life study in 156 osteoporotic patients compared 2 doses of calcifediol—266 µg monthly versus 266 µg every 2 weeks [24]. Both groups received antiresorptive agents. A significant increase in 25OHD concentrations was observed with both regimens. The final level was clearly higher in patients who received the fortnightly regimen (56.2 ± 18.5 ng/mL (140.2 ± 46.1 nmol/L) vs 38.8 ± 12.5 ng/mL (96.8 ± 31.1 nmol/L); *p* < 0.01).

Graeff-Armas et al. [25] conducted a randomized, controlled, double-blind study in 91 patients (53 female and 38 male) with a mean age of 63 years. Four groups were established, namely, 10 µg/day, 15 µg/day, and 20 µg/day of calcifediol, as well as a group receiving 20 µg/day of cholecalciferol. The treatment period was 6 months, with a follow-up of 6 months, during which treatments were discontinued. The 3 groups that received calcifediol had significantly higher concentrations of 25OHD than those treated with cholecalciferol. The increase was linear, with 2 nmol/L/µg of cholecalciferol versus 5 nmol/L/µg in those supplemented with calcifediol. The steady state was reached at approximately 8.5 weeks with constant dosing. The suppression of treatment at 6 months resulted in reduced 25OHD concentrations to baseline values, with the fastest decline in patients who had received calcifediol. The authors reported that an elimination rate lower than that of calcifediol would enable a rapid response to dose adjustments.

In a recent publication by Jódar et al. [26], which was the continuation of the Perez-Castrillón study [16], the authors reported short-term results for patients followed for up to 12 months. From 4 months onward, one of the arms receiving calcifediol was switched to placebo, thus discontinuing the treatment. Suppression of calcifediol treatment returned 25OHD to baseline values after 8 months of suppression, whereas treatment continuation lead to steady state consistent with the findings of Graeff-Armas et al. [25].

Corrado et al. [27] compared monthly doses of cholecalciferol and weekly doses of cholecalciferol and calcifediol in 107 postmenopausal females. Patients were randomized to 4 groups: 3 with cholecalciferol (300,000 IU single dose, 100,000 IU every 2 months, and 7000 IU/week) and 1 with calcifediol 7000 IU/week. Greater increases and improved muscle function were achieved in patients treated with calcifediol than in those treated with cholecalciferol. In patients receiving cholecalciferol, the largest increases were with weekly doses.

In a study of 50 osteopenic or osteoporotic females that compared 2 doses of calcifediol (20 µg/day and 30 µg/day), the peak 25OHD level was observed at 90 days in the first group (59.3 ng/mL, 148 nmol/L) and at 60 days in the second group (72.3 ng/mL, 180.4 nmol/L). Higher doses achieved faster increases. Subsequently, those concentrations were maintained or slightly reduced in both groups [28]. Table 2 shows the main studies.

There is a lack of international consensus on optimal treatment schemes. Vitamin D supplementation is necessary to obtain treatment efficacy and avoid inadequate responses. Consequently, numerous agencies and scientific organizations have developed recommendations for vitamin D supplementation and guidance on optimal serum 25(OH)D concentrations. The bone-centric guidelines recommend a target 25(OH)D concentration of 20 ng/mL (50 nmol/L), and age-dependent daily vitamin D doses of 400–800 IU. The guidelines focused on pleiotropic effects of vitamin D recommend a target 25(OH)D concentration of 30 ng/mL (75 nmol/L), and age-, body weight-, disease-status, and ethnicity-dependent vitamin D doses ranging between 400 and 2000 IU/day [29].

## 4. Safety of Calcifediol

Most studies comparing calcifediol and cholecalciferol are small, and many did not collect safety data. Those that did, analyzed few patients or did not report adverse events related to hypervitaminosis D.

Toxicity is an infrequent event with either calcifediol or cholecalciferol, and blood levels <150 ng/mL (374.3 nmol/L) are considered safe. In fact, treatment with compounds such as calcifediol should only give cause for concern in patients who may be particularly sensitive to them [30]. The main manifestations of toxicity—hypercalcemia and hypercalciuria—are infrequent and result from high doses taken over long periods (i.e., inadequate intake, with doses much higher than those recommended by guidelines or summaries of product characteristics) [31]. Toxicity may also be due to associated diseases that increase the synthesis of 1,25-dihydroxycholecalciferol or genetic defects that alter the metabolism of vitamin D by reducing the concentration of 24,25-dihydroxycholecalciferol, a safety mechanism that prevents toxicity. Furthermore, data from the Spanish Pharmacovigilance system reveal few reports of hypercalcemia or hypervitaminosis D [32].

In the biggest randomized controlled trial [16], comparing the efficacy and safety of calcifediol 0.266-mg soft capsules with cholecalciferol in 303 postmenopausal women, no deaths or serious adverse events were reported, and the maximum 25OHD level recorded was 60 ng/mL (149.7 nmol/L). The authors concluded that calcifediol was an effective and safe option for reaching optimal 25OHD levels in vitamin D-deficient postmenopausal women.

Minisola et al. [8] studied the effects of three therapeutic regimens of calcifediol (20 μg/day, 40 μg/day, and 125 μg/week) on the increase in circulating levels of serum 25OHD after 3 months of treatment. The authors found that 25OHD levels rose significantly, remaining within the safety interval, and that no significant changes were recorded in calcium and phosphate metabolism or bone turnover, thus indicating the compound to be safe, with no need for close monitoring of 25OHD concentrations in the short–medium term.

## 5. Predictability of Calcifediol

Vitamin D serum levels depend not only on intake and skin production, but also on genetic factors that can be modified by epigenetic factors, in addition to phenotypic factors that depend on the individual [6]. Thus, in a significant percentage of patients, food supplementation is not sufficient to maintain adequate levels, especially in institutionalized patients and in winter and spring [10].

In cases of vitamin D deficiency or insufficiency, administration of fortified foods or supplements is necessary. The most widely used supplement, for which most evidence is available, is cholecalciferol, while calcifediol, i.e., 25-hydroxyvitamin D3, the main metabolite for the assessment of vitamin D status, is less used, owing to lack of availability in many countries to date. In both cases, dosage is determinant of efficacy, with higher doses or frequencies generating higher concentrations; however, other factors are also involved and interact in a different way with each drug.

Cholecalciferol is an inactive nutrient that requires a complex metabolism to enable it to bind to the receptor, where it exerts its effect [30]. Several linear regression-based mathematical models have been generated to determine the increase in vitamin D status based on the dose administered. However, these models only explain 37% of the response, since other factors not included in the model may intervene, such as the capacity for absorption, genetic factors, and obesity [31].

Some situations enable greater predictability in the administration of calcifediol than with cholecalciferol, such as obesity, malabsorption, and liver failure. Obese persons have a poorer vitamin D status for several reasons [33], including entrapment of cholecalciferol in excess adipose tissue, leaving a smaller substrate for hydroxylation, along with reduced exposure to sunlight and the inflammatory state associated with obesity [34]. Calcifediol is characterized by greater solubility, decreased entrapment in adipose tissue, a smaller volume of distribution, and a shorter half-life. In addition, the expression of cytochrome P450 (CYP2R1), the main vitamin D 25-hydroxylase, has been shown to be decreased in obesity, in both animals [35], and humans [36]. Absorption of cholecalciferol is mediated by a transport protein, which is transported by chylomicrons, reaching the systemic circulation via the lymphatic system, while calcifediol passes directly into the blood through the portal vein and does not require bile acids or the formation of micelles [37]. This diminishes the effect of obesity and malabsorption on vitamin D status and 25OHD concentrations following administration of calcifediol [38]. Cholecalciferol supplements produce a lower increase in vitamin D status [39]. Charoenngam et al. [40] assessed the effect of the 2 compounds on obesity and malabsorption in 6 patients with malabsorption and 10 healthy individuals and reported interesting pharmacokinetic data. After administration of 900 µg of cholecalciferol, the area under the curve (AUC) was 68% lower for patients with malabsorption than for healthy patients. However, the administration of 900 µg of calcifediol revealed no differences between the 2 groups. The authors then divided the healthy patients into several groups based on body mass index (BMI), and those with a higher BMI who were supplemented with cholecalciferol had a lower AUC than healthy individuals with a normal BMI. Among individuals who received calcifediol, no differences were observed between those with normal and those with elevated BMI.

Given that hydroxylation of cholecalciferol and synthesis of transport protein (DBP) occur in the liver, administration of the hydroxylated metabolite enables a more effective response [41] independently to liver status.

The relationship between 25OHD and related genetic factors has shown little impact. This is not the case for cholecalciferol, with a variable response being observed depending on the polymorphism presented by the patient. The influence of genetic factors is due to enzymes involved in the metabolism of vitamin D, which can be classified into metabolic, catabolic, and receptor enzymes. Metabolic enzymes, including DHCR7, CYP2R1, and DBP, are involved in the synthesis of 25-vitamin D and the transport of its metabolites. Catabolic enzymes include CYP24A1, and receptor enzymes include VDR, a nuclear protein belonging to the nuclear receptors of ligand-activators of transcription factors [9]. All these elements are genetically and epigenetically controlled.

In the case of calcifediol, genetic factors probably play a smaller role, since the metabolite measured is administered directly. The variation is determined by the effect of ultraviolet radiation on the skin or vitamin D administered with diet, thus implying that the administration of calcifediol has predictable effects on the increase in metabolite values, since it depends on the dose administered.

CYP2R1 alleles with decreased 25-hydroxylase activity have been recorded in 3 Caucasian individuals per 1000 and may be responsible for up to 8% of variation in vitamin D administration [42,43]. The percentage increases in areas with greater exposure to sunlight. In other polymorphisms, this percentage has not been calculated.

## 6. Conclusions

Calcifediol is more potent and more rapid than cholecalciferol increasing 25OHD levels, in both the short and the long term. In the long term, differences between the levels of 25OHD achieved by both molecules are reduced, although the effect of calcifediol continues being greater. In addition, the predictability of the response to calcifediol, unlike cholecalciferol, is independent of basal 25OHD levels, and its efficacy is less dependent on other comorbidities, such as obesity or malabsorption, and genetic or/and epigenetic factors.

Data on safety are scarce, although calcifediol has proven to be safe when administered in different regimens and for several months, with a very low risk of toxicity.

## Figures and Tables

**Table 1 nutrients-14-01943-t001:** Short-term studies.

Authors	Type of Study	Population	Design	Baseline Vitamin D ng/L/nmol/L	Methods Vitamin D	Superiority of Calcifediol	Other Data
Russo et al. [12]	Open	18 pre- and postmenopausal females	One arm with 500 μg of 25D_3._ 16 weeks	18.1 ± 12.5 ng/mL 45.1 ± 31.1 nmol/L	RIA	NA	88% > 30 ng/mL (74.8 nmol/L)
Minisola et al. [8]	RCT	87 postmenopausal females	Three arms of 25D3 20μg/day,40μg/day,125μg/week. 16 weeks	16.5 ± 7.5 ng/mL 41.1 ± 18.7 nmol/L	Chemiluminiscence	NA	100% > 30 ng/mL (74.8 nmol/L)
Cashman et al. [13]	RCT	56 adults (25m, 31f) > 50 years	Three arms of 20μg/day D3, 7μg/day and 20μg/day 25D3. 10 weeks	17.4 ± 4.9 ng/mL 43.6 ± 122.3 nmol/L	ELISA	YES	>Dose 20μg/day 25D3
Bischoff-Ferrari et al. [14]	RCT	20 postmenopausal females	Two arms, 20μg/day D3 vs20μg/day 25D3. 16 weeks	13 ± 3.8 ng/mL 32.4 ± 9.4 nmol/L	HPLC-MS/MS	YES	-
Jetter et al. [11]	RCT	35 females aged 50–70 years	7 arms:20μg/day and 140μg/week of D3 vs 20μg/day and 140 μg/week of 25D3 and combination of both arms. 15 weeks	13 ± 5 ng/mL 32.4 ± 12.4 nmol/L	HPLC-MS/MS	YES	Long-term kinetics similar between the two supplements
Shieh et al. [15]	RCT	35 subjects aged >18 years	Two arms 60μg/day of D3 vs 20μg/day of 25D3. 16 weeks	<20ng/ml	HPLC-MS/MS	YES	Determination of free vitamin D with superiority of calcifediol
Perez-Castrillón et al. [16]	RCT	303 postmenopausal females	Two arms 625μg/month D3 vs 266μg/month 25D3. 16 weeks	13 ± 3.9 ng/mL 32.4 ± 9.7 nmol/L	Chemiluminiscence	YES	Greater efficacy at one month and four months for both total vitamin D and free vitamin D

RIA: Radioimmunoassay; HPLC: Liquid chromatography; HPLC-MS/MS: Liquid chromatography coupled to tandem mass spectrometry detection.

**Table 2 nutrients-14-01943-t002:** Long-term studies.

Authors	Type of Study	Population	Design	Baseline Vitamin D ng/mL/nmol/L	Methodology Vitamin D	Superiority of Calcifediol	Other Data
Larrosa et al. [18]	Open	70 subjects (11 males and 59 females	After loading dose (1064 μg 25-D3 in 1 month) Three arms: 266 μg /month,266 μg /3 weeks,266μg /2 weeks. 28 ± 14 months	17.6 ± 6ng/mL 43.9 ± 14.9 nmol/L	RIA	NA	78%, 89%, 93% > 30 ng/mL (74.8 nmol/L) 4%, 11%, 19% > 95 ng/mL (237.1 nmol/L)
Larrosa et al. [19]	Open	129 subjects (109 females, 20 males)	After loading dose (1064 μg 25-D3 in 1 month) Two arms: 20 μg/day D3 vs 266 μg/3 weeks. 12 months	16 ± 5 ng/mL 39.9 ± 12.4 nmol/L	RIA	YES	
Rossini et al. [20]	RCT	271 females	Two arms 21 μg/day D3 vs 100 μg/week. 12 months	22 ± 6 ng/mL 54.9 ± 14.9 nmol/L	RIA	NO	
Navarro-Valverde et al. [21]	RCT	40 postmenopausal females	4 arms:20 μg/day D3 vs20 μg/day, 266 μg/week,266 μg/2 weeks 25-D3. 12 months	15.5 ± 1.7 ng/mL 38.7 ± 4.2 nmol/L	HPLC	YES	Dose dependent effect
Ruggero et al. [22]	RCT	67 subjects (42 females and 25 males)	Two arms: 20 μg/day D3 vs 20 μg/day 25-D3. 7 months	10 (4-16) ng/mL 24.9 (9.9-39.9) nmol/L	RIA	NO	Initial differences but no differences at 210 days
Graeff-Armas et al. [25]	RCT	91 subjects (53 females and 38 males)	Four arms: 20 μg/day D3 vs 10 μg /day,15 μg/day,20 μg/day 25-D3. 6 months	19.2 ± 6.8 ng/mL 48 ± 17 nmol/L	HPLC-MS/MS	YES	Dose dependent effect. Suppression of the supplement reduced vitamin D levels to baseline
Corrado et al. [27]	RCT	160 postmenopausal females	Four arms: 7500 μg single dose, 2500 μg/2 months, 175 μg/week D3 vs 116μg/week 25-D3. 6 months	13.4 ± 4.3 ng/mL 33.4 ± 10.7 nmol/L	Chemiluminescence	YES	Dose dependent effect
Jodar E et al. [26]	RCT	303 postmenopausal females	Two arms 625 μg/month D3 vs 266 μg/month 25D3. 12 months	13 ± 3.9 ng/mL 32.4 ± 9.7 nmol/L	Chemiluminescence	YES	
Gonnelli et al. [28]	RCT	50 osteopenic or osteoporotic females	Two arms, 20 μg/day, 30 μg/day No control with cholecalciferol 6 months	15.6 ± 4.8 ng/mL 39.4 ± 11.9 nmol/L	Chemiluminescence	NA	90 days: 59.3 ng/mL (148 nmol/L) dose 20 μg/day 60 days: 72.3 ng/mL (180.4 nmol/L) dose 30 μg/day

RIA: Radioimmunoassay; HPLC: Liquid chromatography; HPLC-MS/MS: Liquid chromatography coupled to tandem mass spectrometry detection.

## Data Availability

Not applicable.

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
