# Peer review of "Treatment of Vitamin D Deficiency with Calcifediol: Efficacy and Safety Profile and Predictability of Efficacy"

_nutrients, 2022, doi:10.3390/nu14091943_

Round 1

Reviewer 1 Report

Dear Auhtors,

I have read the manuscript and I send you my comments:

1) methods are missing and the type of review is not described

2) the characteristics of each patient in each study is missing

3) the levels of vitamin D before the treatment is missing

4) the results are very similar to the study published in 2021 and reported in references number 16

5) the moethods used to evaluate the levels of vitamin D in each study is missing 

6) how you evaluate the development of adverse drug reactions?

Author Response

Response to reviewer 1.

I have read the manuscript and I send you my comments:

  • Methods are missing and the type of review is not described

It´s added a method section

Methods: We performed a comprehensive review of the literature through MEDLINE, PubMed, Web of Science, Scopus, and Embase electronic databases. Potentially relevant articles were sought by using the search terms in combination as Medical Subject Headings (MeSH) terms and text words: “calcidiol trial”, “cholecalciferoltrial vs calcidiol”, “calcidiol safety “. In addition, we scanned the reference lists of the retrieved publications to identify additional relevant articles. The search was supplemented using the MedLine option 'Related Articles'. No language restrictions were applied. The abstracts for each article were studied to ensure relevance and significance to the review. The safety was evaluated to  by the presence of hypercalcaemia, hypercalciuria and nephrolithiasis or by clinical manifestations derived from the presence of hypercalcaemia and vitamin D levels.

  • The characteristics of each patient in each study is missing

The general characteristics, sex and menopause, are included in the tables and in the text.

3) the levels of vitamin D before the treatment is missing

This values are added in the tables

  • The results are very similar to the study published in 2021 and reported in references number 16

Reference is made to data at 12 months while reference 16 is data at 4 months

5) the moethods used to evaluate the levels of vitamin D in each study is missing 

The methods used to evaluate the levels of vitamin d was included in the tables

6) how you evaluate the development of adverse drug reactions?

The safety was evaluated to  by the presence of hypercalcaemia, hypercalciuria and nephrolithiasis or by clinical manifestations derived from the presence of hypercalcaemia and vitamin D levels.

Reviewer 2 Report

This review covers the relevant research regarding the treatment of vitamin D deficiency with calcifediol. It is a well written manuscript and the authors provide all relevant studies published to date including the studies which compared calcifediol and cholecalciferol. The authors conclude that calcifediol is more potent in increasing 25(OH)D levels in the short and long term. I have the following comments:

  1. In the introduction section the authors should provide a detailed description of the search strategy. Which bibliographic databases were searched for relevant articles? Which terms were used?
  2. What is the half-life of cholecalciferol, calcifediol and calcitriol?
  3. In addition to the concentrations in ng/mL please provide the values in nmol/L.
  4. What doses of vitamin D supplements are generally indicated in postmenopausal women? How about in patients with osteoporosis or osteopenia? This information should be added in the long-term efficacy chapter.

Author Response

Response to reviewer 2.

  1. In the introduction section the authors should provide a detailed description of the search strategy. Which bibliographic databases were searched for relevant articles? Which terms were used?

It´s added a method section We performed a comprehensive review of the literature through MEDLINE, PubMed, Web of Science, Scopus, and Embase electronic databases. Potentially relevant articles were sought by using the search terms in combination as Medical Subject Headings (MeSH) terms and text words: “calcidiol trial”, “cholecalciferoltrial vs calcidiol”, “calcidiol safety “. In addition, we scanned the reference lists of the retrieved publications to identify additional relevant articles. The search was supplemented using the MedLine option 'Related Articles'. No language restrictions were applied. The abstracts for each article were studied to ensure relevance and significance to the review.

  1. What is the half-life of cholecalciferol, calcifediol and calcitriol?

The half life is added

  1. In addition to the concentrations in ng/mL please provide the values in nmol/L.

Both concentrations, ng/ml and nmol/l, were indicated

  1. What doses of vitamin D supplements are generally indicated in postmenopausal women? How about in patients with osteoporosis or osteopenia? This information should be added in the long-term efficacy chapter.

This information was added

Round 2

Reviewer 1 Report

Dear Authors,

I have read again the manuscript and I more data related to the method used to evaluate the development of ADRs must be added